

# Information and decision-making needs of psychiatric patients: the perspective of relatives

Sarah Liebherz, Lisa Tlach, Martin Härter and Jörg Dirmaier

Department of Medical Psychology, University Medical Center Hamburg-Eppendorf, Hamburg, Germany

## ABSTRACT

**Background**. Mental illness may strongly affect relatives' lives. Therefore, it is important to empower relatives by providing health information according to their preferences.

**Methods**. An online cross-sectional survey was conducted using a purpose-designed questionnaire on online health information and decision-support needs.

**Results**. Prevalent reasons for online health information search of the 185 participating relatives were the need for general information and the insufficiency of the information given by the health care provider. The most difficult treatment decisions concerned the treatment setting (inpatient or outpatient) as well as the psychopharmacological treatment.

**Discussion**. Since psychiatric patients' relatives report extensive information and decision-support needs, it is essential to address their needs in health information material. Assessment of relatives' needs when developing health information materials is recommended.

## INTRODUCTION

Mental illness—especially in severe cases—may significantly influence the lives of people in the affected persons' immediate surroundings (e.g., relatives, carers and friends).[1] Living with or caring for a person with a mental illness may increase the relatives' stress level and may cause health problems and a reduced quality of life (*Ali et al., 2014*; *Skundberg-Kletthagen et al., 2014*; *Weimand et al., 2010*). As caring for a mentally ill person may be emotionally challenging and lead to grief, fear and isolation, carers themselves need support (*NICE, 2014*).

To deal with these various challenges, empowerment for relatives through comprehensive information about the affected persons' illness is useful (*Friedrich et al., 2012*). Previous studies on relatives' information needs (mostly focusing on schizophrenia or affective disorders) have confirmed relatives' interest in (written) information about the illness and

Corresponding author
Sarah Liebherz, s.liebherz@uke.de

---

[1]As the majority of the participants of our survey are family members of the affected persons and the cited literature refers mostly to family members, we use the term "relative" here, even if other people from the affected persons' immediate surroundings are also included.
emphasize the need for psychoeducation. However, the demand for information often remains unmet (*Angermeyer, Diaz Ruiz de Zarate & Matschinger, 2000*; *Gantt, Goldstein & Pinsky, 1989*; *Gümüş, 2008*; *Mueser et al., 1992*; *Wancata et al., 2006*; *Wei et al., 2010*). Frequently, relatives do not know the correct diagnosis, do not get adequate information on the current medication or are not able to recognize the symptoms (*Gantt, Goldstein & Pinsky, 1989*; *Gaskill & Cooney, 1991*; *Wittmund, Bischkopf & Angermeyer, 2001*). They report a lack of involvement in the treatment process and often feel alone as they do not know other people being in the same situation (*Gaskill & Cooney, 1991*). Various studies show that unmet information needs of relatives concern medical and psychiatric treatment, medication and side effects, early warning symptoms, sleeping problems, communication with relatives, denial and noncompliance, dealing with common problems, burnout and stress, the health care system, social relationships, coping with bizarre and assaulting behavior, as well as patient rights (*Chien et al., 2003*; *Gümüş, 2008*; *Mueser et al., 1992*; *Pollio, North & Foster, 1998*; *Sung, Hixson & Crofts Yoker, 2004*; *Wancata et al., 2006*; *Wei et al., 2010*). Therefore it seems necessary to inform relatives on these issues. Besides information leaflets, one way to enable and encourage patients and their families and friends to participate in medical decisions is to provide high quality and feasible (web-based) patient decision aids. Patient decision aids are evidence-based tools that support people to deliberate, independently or in collaboration with others, about choices they face by considering relevant attributes of the options (*Elwyn, Frosch & Rollnick, 2009*). In addition to unmet information needs (especially concerning treatment options), patient decision aids may also address unmet decision-support needs.

As a low-threshold opportunity, the Internet bears the potential to deliver interactive, personalized, and individualized content at comparatively low costs to a large number of users at the time, place and learning speed the individual user prefers (*Weymann et al., 2013*). Internet interventions can help to inform about mental disorders, to find local treatment services and to prepare for doctor contacts.

Patient education and patient involvement are relevant components of shared decision-making (SDM). SDM is defined as an interactional process in which the patient and the physician (or another health care provider) aim to reach a decision together that is based on shared information and the best available evidence (*Härter, 2004*). The definition of SDM focuses on the involvement of patient and health care provider. However, relatives are often involved in treatment decisions also—for example to support the patient, both during the medical encounter to provide emotional, informational, or practical support (*Wolff & Roter, 2008*) or when discussing the issue without the health professional. Support of relatives during medical encounters can change the situation, e.g., in influencing the patient's relationship with the physician and in increasing the complexity of the encounter (*Beisecker & Moore, 1994*). A recent review of triadic medical encounters (*Laidsaar-Powell et al., 2013*) found that some companion behavior were helpful (e.g., informational support) while other were less helpful (e.g., dominating or demanding behaviors). Preferences for involvement of others varied widely. The authors give recommendations for health professionals including encouragement/involvement of

accompanying persons, emphasis of helpful companion behaviors as well as clarification of the role preferences of the patient and the accompanying person.

Assessment of service users' and experts' views on the information and decision-support needs of patients with mental disorders is suggested as being one key element in the systematic development of high quality decision aids (*Elwyn et al., 2006*). A recent systematic review shows that research on information and decision-making needs is largely restricted to schizophrenia and depressive disorders (*Tlach et al., 2015*). Moreover, evidence on decision-support needs from the perspective of patients' families and friends is lacking. It can be assumed that relatives can have different roles in the decision-making process, e.g., (1) If the affected person is incapable to make the decision alone (e.g., during a psychotic episode), the relative can make the decision—e.g., as a legal representative; (2) The relative can be an important advisor and can support the affected person to make the decision; (3) The relative can be affected by the decision made by the affected person (e.g., if the affected person takes psychotropic drugs or not); (4) The relative can be faced with decisions concerning their own lives (e.g., getting professional support themselves, keeping in touch with the affected relative during acute illness episodes or not).

Against this background, this paper aims to investigate (online) information needs of psychiatric patients' relatives. Moreover, this paper has a focus on treatment decisions in the treatment course of the affected person and concentrates on the following questions:

1.  Which kind of treatment decisions do the relatives remember?
2.  Which kind of decisions do they remember as difficult?

All relevant treatment decisions remembered by the relatives are explored—regardless of their own role in the decision-making process.

## MATERIALS AND METHODS

An online cross-sectional study using a purposed-designed survey (self-rating) addressing people reporting experience with mental disorders as a relative, friend or acquaintance was employed. The questionnaire was purpose-designed, as there was no questionnaire on mentally ill peoples' relatives' decision-support-needs.

### Survey development

The choice of the topics focused in the survey was based on a systematic review concerning information and decision-making needs (*Tlach et al., 2015*) as well as contact with experts in the field.

The survey was pilot-tested for clarity, ease of use and smooth functioning with 27 participants (research assistants and student assistants). It was also adjusted with different experts in the field of (e-)mental health and SDM research. The items on treatment decisions derived from evidence-based treatment options named in the current treatment guidelines. Since these options differ between different mental health conditions, the questions were tailored depending on the affected persons' mental illness stated. Since the survey was developed as a pre-requisite for the development of web-based patient decision aids, an online sample was examined.

The questionnaire involved five items on sociodemographic characteristics of the participants (age, sex, level of education, partnership status, country of birth), one item concerning the clinical characteristics of the affected person (self-report primary diagnosis) and one item on the relation to the affected person (partner, parent, child, other relative, friend or acquaintance). The main part focused four sections:

1. **Internet use**: three items addressed the frequency of general Internet use, Internet use on general health topics and Internet use on the affected persons' mental disorder on a five-tier scale: (almost) daily, at least once a week, at least once a month, less than once a month, never.

2. **Online health information needs**: two items addressed the reason for information search (six pre-specified answer options) as well as the sort of required information (seven pre-specified answer options). One additional option "other" in both items with a free-text answer option was applied. Multiple answers were allowed. The pre-specified answer options were based on relevant information needs derived from literature search (*Tlach et al., 2015*) as well as contact with experts in the field.

3. **Role in decision-making**: preferred as well as actual role in decision-making with two items adapted from the Control Preference Scale (*Degner, Sloan & Venkatesh, 1997*). The participants had to state (from their point of view) if the affected person or the physician should make/has made the decision alone or together—in accordance with the concept of shared decision-making.

4. **Treatment decisions:** sixteen items concerning evidence-based treatment options for the respective illness—identified through systematic literature (*Tlach et al., 2015*) and guideline search (*DGPPN et al., 2015*; *NICE, 2014*). The items included the following topics: treatment setting, start of treatment, psychopharmacological treatment, psychotherapeutic treatment, combined treatment, alternative treatment. Participants stated if their affected relative ever had made these decisions and (if yes) assessed the difficulty of these decisions (very difficult, rather difficult, rather simple, and very simple). Participants had the possibility to specify other relevant decisions in a free text field introduced by the question: From your point of view, are there other relevant decisions during the course of your relative's illness?

## Data collection

This survey was performed with EFS survey (*QuestBack, 2012*). Participants were recruited online on the e-mental health portal psychenet (http://www.psychenet.de). The portal provides information leaflets, patient decision aids and information on the health care system as a key part. The target group of the portal comprises people affected by mental disorders, their relatives, health care providers as well as other interested people (*Dirmaier et al., 2016*). During the investigation period from January to April 2013, 15.000 visitors per month were registered through web analysis software on the e-mental health portal. The survey was announced at all sub-areas of the portal, introduced by the question: "What kinds of decisions are relevant during the course of a mental disorder?"

Additionally, 48 cooperating self-help groups as well as 10 cooperating hospitals in the area of Hamburg were contacted via e-mail to announce the survey.

Approval for the study was obtained from the ethics committee of the Hamburg Medical Association (Process number: PV4157). All participants were asked for written informed consent. Only participants who gave written informed consent to participate (asked at the beginning of the questionnaire) as well as consent to data use (asked when participants had finished the questionnaire) were included in the analyses.

Participants reporting experience with a relative, friend or acquaintance with a mental illness and being at least 18 years old were included. All data were self-reported.

## Data analysis

Statistical analyses were performed applying the statistical software package PASW Statistics 18 (*SPSS, 2009*). As this research project was explorative, data were primarily evaluated by quantitative descriptive data analysis (absolute and relative frequencies, means and standard deviations). Chi$^2$-tests (exact significance, two-tailed) were applied to test if the preferred role in decision-making (active, shared or passive) or the accordance between the preferred and the actual role, had an influence on the decisional conflict. Answers concerning the items on treatment decisions were dichotomized for this analysis (difficult versus simple decisions). Qualitative data analysis techniques (conventional content analyses) were used to analyze the free text answers concerning information needs and relevant decisions, applying an inductive approach (*Hsieh & Shannon, 2005*; *Mayring, 2000*). This method is considered appropriate when existing theory or research literature on a phenomenon is limited or unclear (*Hsieh & Shannon, 2005*). Responses were categorized into superordinate domains. Responses that included a number of themes were subdivided into various units and separately categorized. The coding was carried out by one student assistant and validated by two members of the research team (SL, LT).

## RESULTS

### Patient flow

During the investigation period from January to April 2013, $n = 437$ relatives, friends or acquaintances of persons with a mental illness started the online-survey. A total of $n = 372$ (85%) gave consent to participate, $n = 200$ (46%) finished the questionnaire and $n = 188$ (43%) participants gave informed consent to data use. Three participants had to be excluded due to implausible data, so the final sample consisted of $n = 185$ participants (42%).

### Sample description

Three quarters of the surveyed (77%) were female. Age ranged from 19 to 82 years (see Table 1). In most cases, participants were parents (35%) or partners (34%) of a person with mental illness. The diagnosis most frequently mentioned was bipolar disorder/mania (45%).

### Internet use

While almost all participants (99%) reported daily or at least weekly Internet use, 46% reported at least weekly internet use for general health information search and 28% reported

**Table 1  Sample description.**

| Characteristics | N | % |
|---|---|---|
| **Sex** | | |
| Female | 143 | 77 |
| Male | 42 | 23 |
| **Age in years** ($M \pm SD$, range) | $48.6 \pm 12.9$, 19–82 | |
| **Education** | | |
| 8–10 years | 53 | 29 |
| More than 10 years | 129 | 70 |
| Other (still at school, without graduation, other) | 3 | 2 |
| **Relationship** | | |
| Living in a permanent relationship | 146 | 79 |
| **Country of birth** | | |
| Germany | 170 | 92 |
| **Affected relative (multiple answers)** | | |
| My partner | 62 | 34 |
| My mother/father | 40 | 22 |
| My daughter/son | 64 | 35 |
| Other relatives | 33 | 18 |
| A friend/acquaintance | 13 | 7 |
| Other (i.e., ex-partner) | 5 | 3 |
| **Main diagnosis** | | |
| Alcohol misuse or additction | 3 | 2 |
| Schizophrenia/psychosis | 38 | 21 |
| Bipolar disorder/mania | 84 | 45 |
| Depression | 27 | 15 |
| Anxiety disorder | 5 | 3 |
| Somatoform disorder | 6 | 3 |
| Eating disorder | 2 | 1 |
| Other | 14 | 8 |
| I don't know | 6 | 3 |

at least weekly Internet use for information search specifically on their relatives' mental illness (see Fig. 1).

## Reasons for online health information search

The most prevalent reasons for online health information search were the need for general information on the relatives' mental disorder (67%) and the insufficiency of the health care provider's information (43%; see Fig. 2).

### Other reasons for online health information search (free-text answers)

Other reasons were the search for support for relatives (for example self-help groups), the search for treatment possibilities—other than medication—, the feeling of being unable to cope, the wish to feel sympathetic to the relative and the search for proves that the diagnoses exists—because other family members refused to recognize this.
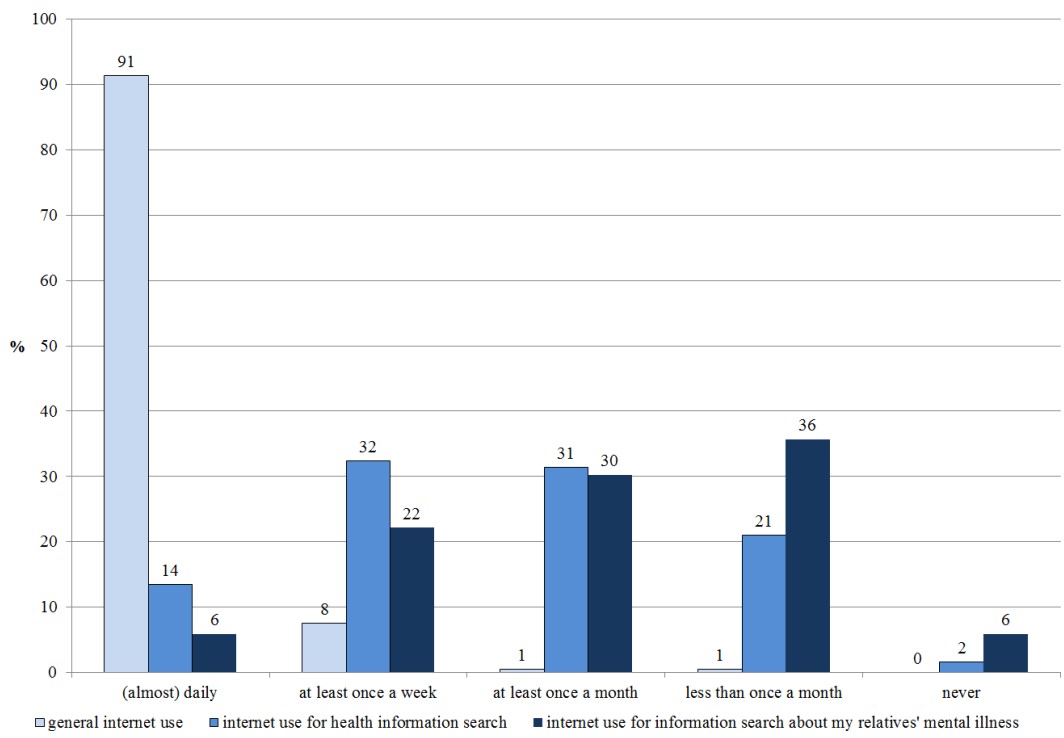

**Figure 1   Internet use ($N = 185$).**

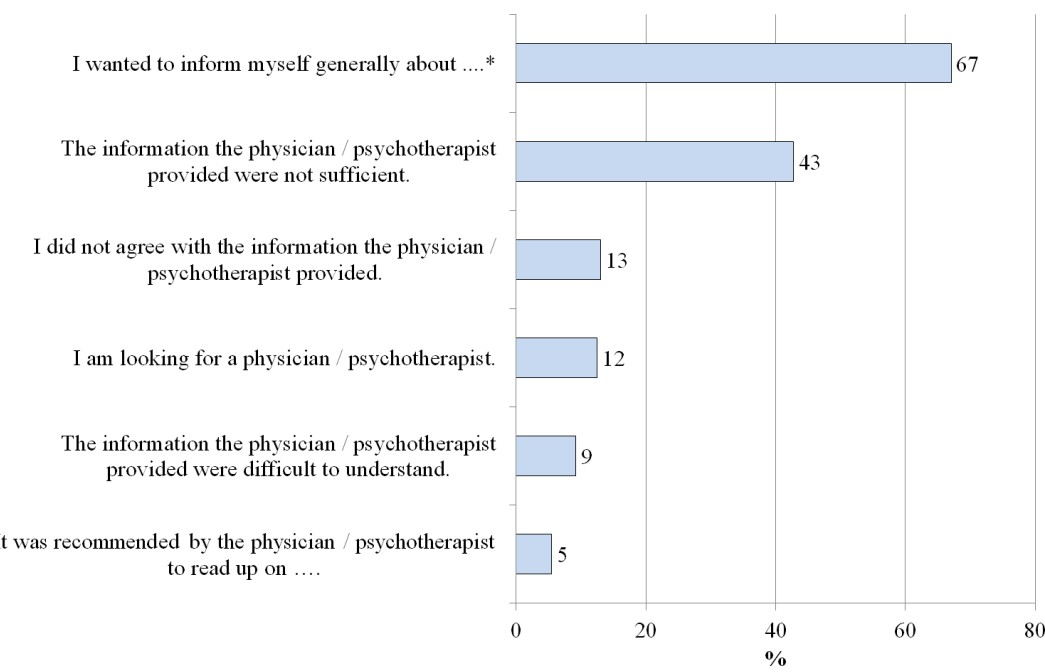

**Figure 2   Reasons for searching the internet ($N = 185$).** * The relative's mental illness previously mentioned was named here.
**Table 2** Online health information needs (*N* = 185).

| | N | % |
|---|---|---|
| Information for relatives | 130 | 70 |
| General information on …[a] (symptoms, causes, course of the disease) | 125 | 68 |
| Tips on dealing with the disease (e.g., coping with everyday life, self-help) | 115 | 62 |
| Information about treatment options (e.g., psychotherapy, drugs) | 111 | 60 |
| Information about chances, risks, and side effects of the treatment options | 99 | 54 |
| Information about self-help groups/exchange with affected/reported experiences of those affected | 80 | 43 |
| Information on psychotherapists, physicians, clinics (e.g., Where can I get treatment?) | 46 | 25 |

**Notes.**
[a]The relative's mental illness previously mentioned was named here.

## Online health information needs

More than a half of all participants mentioned the need for information tailored to relatives (70%), general information on the affected relative's disease (68%), tips on dealing with the disease (62%), information about treatment options (60%) as well as information about chances, risks and side effects of the treatment options (54%; see Table 2).

### Other online health information needs (free-text answers)

In the free-text fields, the participants also stated the need for information on protection of the affected person's children, books or films, the current state of research, heredity as well as civil rights.

## Role in decision-making

Two thirds (65%) of the participants would prefer a shared decision of the affected person and his/her health care provider, while only about one third (38%) remembered the last decisions as shared ones. In total, 48% of the participants experienced a role according to their preferences, while 52% did not. There were more decisions made by the affected person or the physician/psychotherapist alone than preferred (see Fig. 3).

## Treatment decisions and decisional conflicts

Relatives had to state for each treatment decision, if the affected person ever made this decision and if they remembered the decision as difficult or not. The most difficult treatment decisions (remembered as 'rather difficult' or 'very difficult' by more than half of the participants) concerned the treatment setting (inpatient or outpatient) as well as the psychopharmacological treatment. Decisions on taking psychotropic drugs, getting outpatient or inpatient treatment and starting psychotherapy were remembered as 'already made' by at least 80 per cent of all participants. The least common decisions (50% or more did not meet this decision yet) are: psychotherapy OR psychotropic drugs, the application of alternative treatments, the decision to work through a self-help book or not and the decision if the affected relative should quit the current psychotherapy (see Table 3).

The difficulty of making a decision as a percentage of those who had actually made the decision is described in the supplemental material (Fig. S1). However, the decisions remembered as most difficult, remain almost the same.

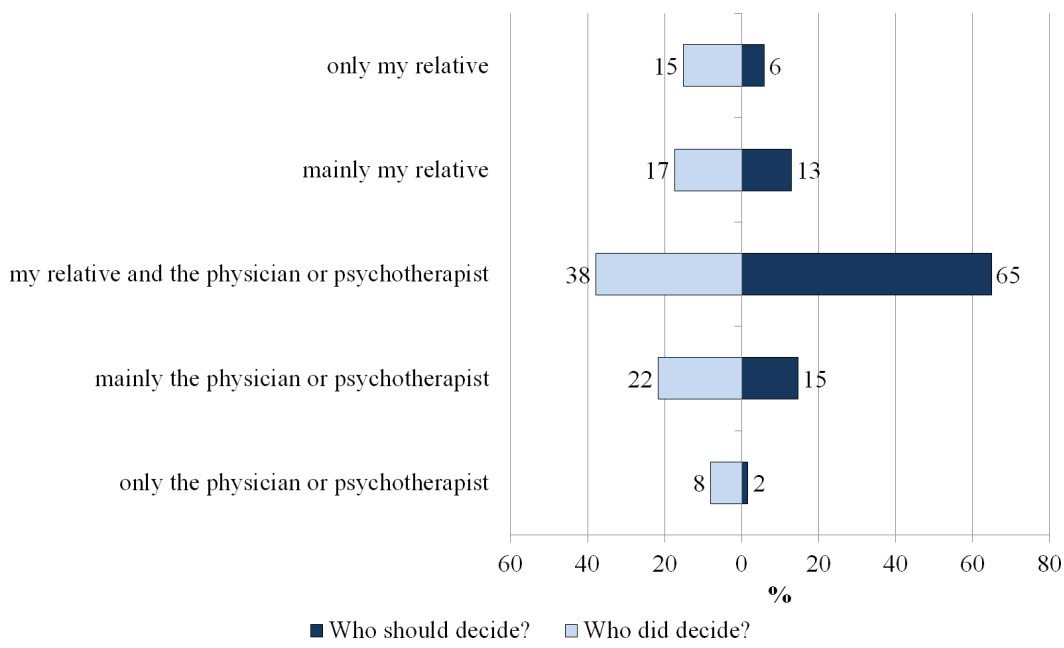

**Figure 3** Preferred and experienced decision-making role (in %, N = 185).

**Table 3** Treatment decisions concerning the affected person (N = 185), figures in %.

| | Decision classified as 'rather difficult' or 'very difficult' | Decision classified as 'rather simple' or 'very simple' | Not made this decision yet |
|---|---|---|---|
| Outpatient or inpatient treatment | 70 | 17 | 13 |
| Taking psychotropic drugs or not | 58 | 29 | 12 |
| Continuing to take a drug or depose again | 57 | 19 | 24 |
| Taking another medication or a different dose | 49 | 23 | 29 |
| Starting a psychotherapy or not | 48 | 34 | 18 |
| Which psychotropic drugs to take | 48 | 30 | 22 |
| Attending a behavioral, psychodynamic or analytic psychotherapy | 37 | 19 | 44 |
| Starting a psychotherapy in addition to psychopharmacological treatment or not | 37 | 26 | 37 |
| Taking psychotropic drugs in addition to the ongoing psychotherapy or not | 37 | 22 | 42 |
| Quit the current psychotherapy or not | 32 | 16 | 53 |
| Doing physical training or not | 24 | 45 | 30 |
| Attending a psychotherapy OR taking psychotropic drugs | 23 | 16 | 62 |
| Working through a self-help book or not | 20 | 25 | 55 |
| Making use of alternative medical services (e.g., herbal medicines) or not | 19 | 21 | 60 |

### *Role in decision-making and decisional conflicts*

There were no significant differences in the perceived difficulty of any treatment decision depending on the preferred decision-making role (active, shared or passive) or the accordance between the preferred and the actual role, except concerning one item: More participants reporting an accordance between the preferred and actual role, remembered the decision 'starting a psychotherapy in addition to psychopharmacological treatment or not' as difficult (see Tables S1 and S2).

### *Other relevant decisions (free-text answers)*

Other decisions relevant for relatives during the course of disease of the affected person focused on dealing with the disease (e.g., informing other relatives or the employer about the mental illness, noticing that the illness restarts, dealing with a lack of illness insight), treatment (e.g., finding the best hospital, physician or psychotherapist, relying on the health care system, aftercare, compulsory hospitalization), being included in the affected persons' life and self-care (e.g., being involved in the relative's life, protecting themselves and other relatives, getting support and being able to stand the strain, keeping in touch with the affected relative during acute illness episodes), lifestyle changes, and life planning (e.g., need for a legal supervision, housing situation, separation from the affected partner, kind and extent of occupational activity, applying for a disabled person's pass, giving up one's own job).

## DISCUSSION

The information and decision-support needs of psychiatric patients' relatives were studied by including 185 participants in an online cross-sectional survey. To our knowledge, this is the first survey on online health information and decision-support needs of psychiatric patients' relatives. Most participants were close relatives of an affected person (partner, parent or child). Concordant with previous studies (*Angermeyer, Diaz Ruiz de Zarate & Matschinger, 2000*; *Chien et al., 2003*; *Gaskill & Cooney, 1991*; *Gümüş, 2008*; *Mueser et al., 1992*; *Sung, Hixson & Crofts Yoker, 2004*; *Wancata et al., 2006*), the participants reported a strong need for general information on the affected person's mental disorder (e.g., symptoms and treatment options), for tips on dealing with the disease and for information tailored specifically to relatives. They also mention the insufficiency of the health care provider's information as main reason for online health information search.

In comparison, samples with participants affected from bipolar disorder/mania, unipolar depression or anxiety disorders (*Liebherz et al., 2015a*; *Liebherz et al., 2015b*) mentioned general information on the disease, information about treatment options as well as tips on dealing with the disease as main interest.

Concerning treatment decisions, decisions on the treatment setting (inpatient or outpatient) and on pharmacological treatment were remembered as most difficult. These decision-making topics are consistent with those mentioned by a sample of patients with affective disorders (*Liebherz et al., 2015b*): The three most difficult decisions overlapped in these two samples. However, the percentage of people remembering these decisions as ''difficult'' was lower in the patient sample than in the relatives sample, especially

concerning the decision on inpatient versus outpatient treatment (53 versus 70%). A sample of patients with anxiety disorders (*Liebherz et al., 2015a*) reported an overall lower percentage of decisions remembered as "rather or very difficult", the three most difficult decisions were " starting a psychotherapy or not" (33%), "inpatient or outpatient treatment" (32%) and "taking psychotropic drugs or not" (30%).

The decision between inpatient and outpatient treatment may be particularly relevant in Germany, where specific psychotherapeutic inpatient treatment settings are a common part of the health care system. In other countries, there are mainly psychiatric facilities whereas in Germany there are also rehabilitation clinics as well as acute hospitals focusing a psychosomatic respectively psychotherapeutic approach (*Schulz et al., 2011*). In current treatment guidelines (*Bandelow et al., 2014*; *DGBS & DGPPN, 2013*; *DGPPN et al., 2015*), inpatient treatment is recommended when outpatient treatment is considered as insufficient (e.g., in case of suicidal tendency, profound psychosocial barriers or resistance to outpatient psychotherapy). Unfortunately, there is lack of evidence concerning the choice of the best treatment setting. To give evidence-based recommendations, randomized controlled trials (RCTs) comparing inpatient to outpatient treatment are required. However, the application of RCTs is difficult in this context—as an allocation to a treatment condition of lower intensity would be considered unethical for severely disturbed patients (*Liebherz & Rabung, 2014*). Thus, the difficulty to meet the decision on inpatient versus outpatient treatment may result from this general lack of evidence or the lack of sufficient information from the health care provider. Moreover, relatives may have experienced a conflict between considering an inpatient treatment as necessary on the one hand and the lack of illness insight of the affected person on the other hand. Relatives may feel overcharged with the affected person's symptoms (e.g., aggression, psychotic symptoms) or medication noncompliance (*Gerson & Rose, 2012*), but they also may be unpleasant with the option of compulsory hospitalization. In the free-text answers, the participants often mentioned related problems like dealing with lack of illness insight or keeping in touch with the affected person in severe illness episodes. To address the relatives' information needs as detailed as possible, further research is needed on the specific difficulties of this decision and on appropriate support possibilities to facilitate this decision.

Further research is also required to clarify the reasons for the difficulty of decisions concerning medication. They may result from reasons such as insufficiency of health care providers' education on psychopharmacological treatment, general mistrust in (psychopharmacological) medication (*Angermeyer & Matschinger, 2004*) or the difficulty in weighing the pros (e.g., response) and cons (e.g., side effects).

The results of this survey confirm the importance to address the relatives' needs in mental health care. These findings are, for example, concordant with the current NICE-guideline for schizophrenia and psychosis, which added some special recommendations in the revised version of 2014 (*NICE, 2014*). The guideline proposes health care providers to work in partnerships with the affected persons and their carers (often family members) and furthermore recommends the assessment of carers' needs and the development of plans to address these needs. Moreover, written and verbal information for carers is recommended, including information on diagnosis, management, outcomes, recovery, support for carers,

roles in the health care system and getting help in a crisis. The guideline also addresses the aspect of collaboration and confidentiality and suggests negotiating these issues as soon as possible. Individual needs as well as pros and cons of information sharing about the affected person should be taken into account. Moreover, the guideline recommends including carers in the decision-making process if the affected person agrees.

The information and decision-support needs of the relatives (e.g., most relevant decision-making topics and high need of general information on the respective disorder) overlap with the affected peoples' needs (*Liebherz et al., 2015a*; *Liebherz et al., 2015b*). Previous studies also found that some information needs are overlapping (e.g., the need for general information on the disease and information on coping with symptoms) whereas others are not (e.g., relatives reported higher information needs concerning communication and social relationships and concerning finding a job and employment status) (*Gümüş, 2008*; *Pollio, North & Foster, 1998*; *Sung, Hixson & Crofts Yoker, 2004*). When designing patient information material, patient decision aids or psychoeducational programs, it is therefore recommendable to include the affected persons', the relatives' as well as the health care provider's ideas and needs.

To provide health information, different media (e.g., information leaflets, written online information, videos or podcasts) may be applied. However, exploring a preference for special media was not part of this survey.

## Limitations

In this study, it was asked for decisions the affected person had to make and if the relative remembered it as difficult. Thus, these two points of view may have been mixed here and we do not know if the decision was difficult for the affected person, the relative or both. Information and decisional needs and conflicts may therefore result from lack of information through the health care provider or from disagreement or communication problems between the affected person and the relative.

Overall, as relatives had to provide information on the affected person's illness and associated treatment decisions, this survey has to deal with recall bias as all information provided are based on the relatives' memory.

Since most of the previous research on relatives' needs focused on relatives of patients with affective disorders or schizophrenia, this study aimed at including other psychiatric patients' relatives. Still, the percentage of relatives of patients with other disorders is low (16%). Furthermore, diagnostic validity is restricted due to self-reported diagnoses. Since the number of participants in some diagnostic subgroups is low, subgroups analyses were not feasible. There is only one study comparing the needs of relatives of patients with affective disorders to relatives of patients with schizophrenia (*Mueser et al., 1992*). Their needs differed only marginally and only concerning the specific symptoms of the particular disease. Further research on the potential differences between different groups of relatives is therefore recommended.

Additional limitations comprise the relatively small sample size as well as the restriction to one country. In other health care systems and cultures, other information and decision-making needs may be primarily focused.

The recruitment of the participants via an online health information portal suggests that these participants were more likely to have unmet information or decision-making needs, were more (online) health information seeking and perhaps also more likely to prefer an active role in the decision-making process. Thus, the information and decision-support needs of these relatives may not be generalizable.

To our knowledge, this is the first study on psychiatric patients' relatives' decision-making needs. Thus, and also because of the first application of the purpose-designed questionnaire, the results are exploratory and need to be confirmed in further studies.

## CONCLUSIONS

Since psychiatric patients' relatives report extensive information and decision-support needs, it is essential to address their needs in health information material (e.g., information leaflets and patient decision aids). Online information appears to be a good opportunity to educate psychiatric patients' relatives on symptoms and treatment possibilities. Relatives are able to access online information autonomously—without the affected relative's or the health care professional's support. Still, relatives may feel that they are taken seriously if health care providers also have a focus on their concerns. Their needs may differ from the affected person's needs with regard to some issues. Therefore, their special interests should be considered when designing and evaluating health information material. Additionally, their needs should be addressed in clinical practice.

## ACKNOWLEDGEMENTS

*psychenet* is a project network supported by the German Federal Ministry of Education and Research (funding code 01KQ1002B) in the region of Hamburg which consists of more than 100 scientific and medical institutions, counseling centers, the Senate and the Chamber of Commerce of the Free and Hanseatic City of Hamburg, companies, as well as patients' and relatives' associations (2011–2015). The vision of the project is to promote mental health today and in the future, concerning early diagnosis and effective treatment of mental illnesses. For more information and a list of all partners, please visit http://www.psychenet.de. The authors would like to thank Iris Extra for her support in data analysis of the free text fields.

### Funding

psychenet is a project network funded by the German Federal Ministry of Education and Research (funding code 01KQ1002B). The funders had no role in study design, data collection and analysis, decision to publish, or preparation of the manuscript.

### Grant Disclosures

The following grant information was disclosed by the authors:
German Federal Ministry of Education and Research: 01KQ1002B.

## Competing Interests

The authors declare there are no competing interests.

## Author Contributions

- Sarah Liebherz conceived and designed the experiments, performed the experiments, analyzed the data, contributed reagents/materials/analysis tools, wrote the paper, prepared figures and/or tables, reviewed drafts of the paper.
- Lisa Tlach conceived and designed the experiments, performed the experiments, analyzed the data, contributed reagents/materials/analysis tools, wrote the paper, reviewed drafts of the paper.
- Martin Härter and Jörg Dirmaier conceived and designed the experiments, contributed reagents/materials/analysis tools, wrote the paper, reviewed drafts of the paper.

## Human Ethics

The following information was supplied relating to ethical approvals (i.e., approving body and any reference numbers):

Approval for the study was obtained from the ethics committee of the Hamburg Medical Association (Process number: PV4157). All participants were asked for written informed consent. Only participants who gave written informed consent to participate (asked at the beginning of the questionnaire) as well as consent to data use (asked when participants had finished the questionnaire) were included in the analyses.

## Data Availability

The raw data has been supplied as a Supplementary File.

## Supplemental Information

Supplemental information for this article can be found online at http://dx.doi.org/10.7717/peerj.3378#supplemental-information.

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
