# Peer review of "Information and decision-making needs of psychiatric patients: the perspective of relatives"

_PeerJ, doi:10.7717/peerj.3378_

## Round 0.1 · original submission · Minor Revisions

The authors should address the detailed feedback of the reviewers when revising their manuscript. Please note that PeerJ does not offer copyediting, so ensure that the English language in the revised manuscript meets our standards.

Reviewer 1 ·

Basic reporting

Introduction

Line 40: Person with a mental illness is perhaps the more accepted terminology to replace “mentally ill person” in this section.

Line 41: It is not clear what is meant here by “health restrictions”, can the authors please elaborate or use an alternative, less ambiguous phrase.

Lines 59-61: The introduction of patient decision-aids seems to be somewhat abrupt and lacking in context. The authors need to provide a clearer segue linking their discussion of unmet information needs and the use of patient decision-aids (which may address unmet information needs via increased knowledge of treatment options but go further to address unmet decision-support needs).

Further, in this same section the authors present a comprehensive review of previous literature relevant to the (unmet) information needs of the relatives of people affected by mental illness. Given that the current study does the same it is important that the authors delineate more clearly the knowledge gaps and how their study addresses these.

Lines 75-77: In discussing the involvement of relatives in treatment decisions, it might be useful to refer to some of the literature on triadic decision-making (doctor-patient-family) and the ways in which family members are involved in and may support (or even challenge) the decision-making process. See for example: Laidsaar-Powell et al., 2013. PEC.

Lines 83-84: With regards to the Tlach et al., 2015 article reference, the authors should clarify if possible whether “affective disorders” encompasses both depressive and bipolar-related disorders.

Lines 85-86: During a psychotic episode, may be more conventional and less ambiguous wording for “loss of reality”.
Lines 84-92: This section relating to the roles played by relatives in treatment decision-making appears to be lacking the related references. Please insert.
Lines 94-99: The authors’ use of personal pronouns in the aims (and later methods) is somewhat at odds with formal academic writing styles. It is suggested that the authors rephrase these sentences in the impersonal voice.

Lines 98-99: It is not necessary to for the authors to state cross-sectional survey as method of data collection in the introduction, especially as this information is restated in the first line of the Methods section.

Method

Lines 119-120: It is unclear what is meant by: “six respectively seven…answer options”

Results

Lines 83-92: Although frequencies/percentages are reported in the related table and figures, it would be good to include these in text also.

Lines 194-207: Even though the free text responses were analysed qualitatively, it would be informative for the reader to know how salient these responses were in the data.

Experimental design

Abstract

Line 27: In the Methods section it is suggested that the authors state here whether the questionnaires were purpose-designed for the study or whether they contained validated measures (or both).

Method –
Line 106: It would be good for the authors to state elaborate on the sentence here to state that the questionnaire they developed included both purpose-designed and validated measures.

Line 111: The authors state that patient clinical characteristics were collected. In addition to diagnosis, did the questionnaire ask for any other clinical characteristics (e.g., time since diagnosis), as this would likely impact on the unmet information needs of relatives (e.g., recently diagnosed versus longstanding diagnosis).

Lines 119-120: It is unclear how the authors generated the 6-7 prespecified answer options on online health information needs? Were these derived from a literature search, consultation with expert working party, pilot study etc?

Lines 122-126: The authors assessed the patient’s experienced role in decision-making (as rated by relatives). Given the potential for recall bias, reliance on patient-report (for consultations where the relative had not attended) and potential for variability in patient’s experienced decision-making role (e.g., due to severity of, fluctuation in mental health symptoms), did the authors limit how long ago or when this treatment decision need to have occurred?
Further, in light of the fact that this survey focused on relatives’ needs, what did the authors choose to not also assess relatives’ preferences for their own involvement/role in decision-making as opposed to just the roles of the patient and physician?

Lines 163-164: In addition to the criteria stated here, were there any other eligibility criteria for participation?

Data analysis – lines 167-174: Given the somewhat heterogeneous sample and rationale for individualised, targeted information provision, did the authors consider exploring potential differences in information needs according to different sample characteristics/subgroups (e.g., relatives and friends/acquaintances; preferences for active versus non-active patient role in decision-making)? Later the authors state that analyses according to diagnostic subgroup were not possible due to low numbers in some groups, however, other sample characteristics show a reasonable distribution.

Further, it would be helpful if the authors elaborated on the specific qualitative methods they employed and provide a relevant reference/s.

Validity of the findings

Discussion

Discussion and interpretation of findings is consistent with results; and the conclusions are appropriate.

Lines 292-315: In this section, the authors acknowledge some of the limitations of the present research (e.g., reliance on self-reported diagnosis and fact that subgroup analyses according to diagnostic group were not possible). However, another noteworthy limitation to acknowledge is that the sample was recruited through an online health information portal and as such it could be that these relatives were more likely to have unmet information/decision-support needs, be more (online) information seeking, and be more likely to prefer an active patient role in decision-making (given that information gathering is a component of SDM). Thus the information and decision-support needs of these relatives may not be generalisable to this population more broadly.

Lines 317-319: As well as commenting on the overlap between the information and decision-support needs of patients and their relatives (Liebherz et al., 2015a; 2015b), it would be worthwhile also pointing out any observed differences in their reported needs. This seems justified in light of the other cited research cited in this section as well as relatives’ reported need for information that is tailored specifically to them.

Additional comments

This study presents cross-sectional survey findings on the unmet informational and decision-support needs of the relatives of people affected by a mental illness. Despite relatives’ acknowledged involvement in and influence on treatment decision-making both in mental and mental health settings, relatives have been largely understudied and underserved in terms of information and decision-support. Findings elucidate important targets for future information interventions for this group.

Although generally well-written, I feel that the manuscript would benefit from English language editing as a number of typographic and grammatical errors were identified as well as several instances of ambiguous or unconventional wording.

·

Basic reporting

The manuscript is well-structured and referenced, and there is sufficient detail There is some changing between past, present, and future tense throughout the paper (e.g. lines 106-110, ‘involved’ and then ‘focuses’ (on). The language is mostly understandable/acceptable but there are some other places where editing is needed.

Experimental design

The aims could be described a little more clearly in the last paragraph of the introduction. Could you move the description of how the survey was developed and tested to the start of the Survey Development section? Also some more description of the development (not just the testing) would be useful. E.g. how did you decide which topics to cover? Did you speak with carers, health professionals etc?

Validity of the findings

The data are sound and the analysis, while simple, is suited to the study's purpose. Could you look at whether greater discrepancy between the person’s desired and actual role in decision making has an impact on the kind of information needs they have?

I’m wondering if you should also report difficulty of making a decision as a percentage of those who had actually made the decision. If they’ve not made the decision yet, it looks as if this reduces the percentage who found the decision difficult, i.e. those decisions not made yet look as if they’re easier, but in actual fact just haven’t been made.

Additional comments

Thanks for this interesting paper. I think this is an important topic for consumers. Just a few comments below:

Line 116 – it’s not clear how many options there were (six or seven?).

Role in decision making, lines 119-123 – do you mean their preferred vs actual role in decision-making? This para is a little unclear.

Line 136 – this sentence is unclear.

Table 2 – needs a new title. Some of the phrases in brackets seem to be unfinished. By partnership, do you mean in a relationship?

Line 210 – I’m not sure what the authors mean by ‘respective treatment decision’. Do you mean each decision?

I’d like to see better reporting of the qualitative data. I think it starts on line 221 but it’s not clear that these are the themes and categories.

You’ve mentioned written information. Did you ask respondents about what format they’d like to receive information in? What about videos, podcasts, etc?

The discussion needs a limitations section, e.g. that the survey was conducted in one city/country and some findings may be less relevant to other settings, small sample size, potential sampling bias.

---

## Round 0.2 · accepted · Accept

The authors have addressed all reviewer comments and the revised manuscript is suitable for publication in PeerJ.

Reviewer 1 ·

Basic reporting

No comment

Experimental design

No comment

Validity of the findings

No comment

Additional comments

The authors have provided a thoughtful and comprehensive rebuttal. The have made a number of changes and additions to the original manuscript based on my suggestions, which I belief have strengthened the manuscript considerably.